# Phytopathogenic *Curtobacterium flaccumfaciens* Strains Circulating on Leguminous Plants, Alternative Hosts and Weeds in Russia

**DOI:** 10.3390/plants13050667

**Published:** 2024-02-28

**Authors:** Anna D. Tokmakova, Rashit I. Tarakanov, Anna A. Lukianova, Peter V. Evseev, Lyubov V. Dorofeeva, Alexander N. Ignatov, Fevzi S.-U. Dzhalilov, Sergei A. Subbotin, Konstantin A. Miroshnikov

**Affiliations:** 1Shemyakin-Ovchinnikov Institute of Bioorganic Chemistry, Russian Academy of Sciences, Miklukho-Maklaya Str. 16/10, Moscow 117997, Russia; anna.zem@mail.ru (A.D.T.); a.al.lukianova@gmail.com (A.A.L.); petevseev@gmail.com (P.V.E.); 2Department of Plant Protection, Russian State Agrarian University—Moscow Timiryazev Agricultural Academy, Timiryazevskaya Str. 49, Moscow 127434, Russia; tarakanov.rashit@mail.ru (R.I.T.); dzhalilov@rgau-msha.ru (F.S.-U.D.); 3All-Russian Collection of Microorganisms (VKM), Skryabin Institute of Biochemistry and Physiology of Microorganisms, Pushchino Scientific Center for Biological Research, Russian Academy of Sciences, Prosp. Nauki 5, Pushchino 142290, Russia; dekabr28@rambler.ru; 4Agrobiotechnology Department, Agrarian and Technological Institute, RUDN University, Miklukho-Maklaya Str., 6, Moscow 117198, Russia; an.ignatov@gmail.com; 5Center of Parasitology, Severtsov Institute of Ecology and Evolution, Leninsky Prosp., 33, Moscow 119071, Russia; sergei.a.subbotin@gmail.com; 6California Department of Food and Agriculture, 1220 N. Str., Sacramento, CA 95832, USA

**Keywords:** *Curtobacterium*, soybean, common bean, bacterial wilt, tan spot, genetic diversity, phylogeny

## Abstract

Many bacterial plant pathogens have a broad host range important for their life cycle. Alternate hosts from plant families other than the main (primary) host support the survival and dissemination of the pathogen population even in absence of main host plants. Metabolic peculiarities of main and alternative host plants can affect genetic diversity within and between the pathogen populations isolated from those plants. Strains of Gram-positive bacterium *Curtobacterium flaccumfaciens* were identified as being causal agents of bacterial spot and wilt diseases on leguminous plants, and other crop and weed plants, collected in different regions of Russia. Their biochemical properties and susceptibility to copper compounds have been found to be relatively uniform. According to conventional PCR assays, all of the isolates studied were categorised as pathovar *Curtobacterim flaccumfaciens* pv. *flaccumfaciens*, a pathogen of legumes. However, the strains demonstrated a substantial diversity in terms of virulence on several tested host plants and different phylogenetic relationships were revealed by BOX-PCR and alanine synthase gene (*alaS*) sequencing.

## 1. Introduction

The production of soybeans in Russia increased from 3.4 million tons in 2020 to 4.6 million tons in 2022, and their cultivation area expanded from 2 million hectares to 3 million hectares over the same time period [1]. The production of other legume crops has increased in a similar way. Legume bacterial diseases cause yield losses of up to 50%, and dramatically impact economic revenues and food security [2,3]. Therefore, the development of innovative approaches for the early detection and prevention of possible bacterial infections in legume crops is a task of high priority. One of the most important pathogens threatening legumes is Gram-positive *Curtobacterium flaccumfaciens* [4], which causes bacterial wilt and tan spot. The pathogen has many variants that cause damage to legume crops worldwide by causing leaf, or systemic, disease in plants [5].

*Curtobacterium flaccumfaciens* has been identified as the reason for the devastating outbreak of the bacterial rot of navy beans in 1921, in South Dakota, USA [6]. Since then, this bacterium has been identified worldwide as the cause of diseases of several cultivated plants. The pathogen can penetrate plants through the stomata and hydathodes, and at sites of mechanical damage. It spreads through the vascular system of the infected plant, blocking water movement [7]. As a result, the plant wilts and brown spots or a blight appear on the leaves [5]. Symptoms are especially pronounced in young plants.

Pathovar *Curtobacterium flaccumfaciens* pv. *flaccumfaciens* (Cff) is considered to be specific for leguminous plants, causing severe losses during the cultivation of crops and the storage of yield [7]. The European and Mediterranean Plant Protection Organisation (EPPO) has included Cff in the A2 list of quarantine organisms. Similar measures have also been taken by the Inter-African Plant Protection Council and the Caribbean Plant Protection Commission [8].

This issue has become particularly important for Russia, due to the country’s increasing export of legumes to countries that classify Cff as a quarantine organism. No substantial outbreaks of Cff-based tan spot of legumes have been reported in Russia, but this pathogen has been detected on soybean [9] and common bean plants [10]. Preliminary surveys have also reported the occurrence of Cff as an infective agent of other plants, in particular sunflower [11]. Since Cff is not regarded as a quarantine pathogen in Russia, the spatial distribution of this bacterium has been poorly studied.

European authorities (situation by 2018) accept that Cff was present in Turkey, Serbia and Iran, but they do not list Cff as being present in the countries bordering Russia [12]. There were several reports on its occurrence in Ukraine in the period 2013–2016 [13,14]. Other pathovars of *C. flaccumfaciens* are often reported in China [15], and Cff has been found in the roots of Chinese cabbage [16]. Cff survival has been confirmed in the phyllosphere and in the rhizosphere of many weeds in field conditions [17]. Thus, aspects of the distribution and life cycle of Cff remain unclear.

The goal of this work is the assessment of the diversity of *Curtobacterium flaccumfaciens* strains isolated in Russia from cultivated and weed plants. The study included biochemical tests recommended in the relevant EPPO bulletin and a characterization of the genetic diversity of isolated strains. Their virulence was tested in the context of the artificial infection of soybean and common bean plants, and susceptibility to bactericide was estimated. This multiphasic approach has been proposed as being an appropriate method of integrating different types of information.

## 2. Results

### 2.1. Isolation and Phenotypic Characteristics of Bacterial Strains

In the course of this work, 33 samples of bacterial plant pathogens, putatively related to *Curtobacterium* sp., were isolated. Infected plants with typical symptoms of bacterial spot, blight and bacterial wilt included legumes, as well as other crop and weed plants. These samples formed a collection of strains, which were identified using a continuous numbering system: C001–C033 etc. For reference, the collection included several strains regarded as Cff, provided by All-Russian Collection of Microorganisms (VKM) (Table 1).

On SSM medium (Figure 1B), as well as on NBY medium, most strains formed yellow or light-yellow colonies, which is consistent with the observed dominance of yellow-coloured strains of *Curtobacterium* sp. in other regions [5]. On MSCFF medium, zones of proteolysis are visible, consistent with the characteristic colony appearance expected from Cff (Figure 1A).

The attribution of novel isolates to *Curtobacterium* sp. was initially assessed using the genus-specific diagnostic PCR test [18], as well as using the Cff-specific assay [19]. For strains of the collection that demonstrated a positive signal in both cases, further study was conducted aimed at assessing the physiological and biochemical properties of the isolates. In addition, their virulence to leguminous plants and aggressiveness towards soybean plants were estimated.

All studied strains assigned as Cff were Gram-positive and positive for catalase, and negative for urease, oxidase and indole, as expected from EPPO Bulletin PM 7/100 [20]. In addition, all strains demonstrated caseinase activity, forming zones of hydrolysis on the MSCFF medium but the gelatinase activity of the strains varied. All studied strains were capable of oxidising glucose and mannose, while the ability to oxidise maltose, mannitol and inositol varied. All strains were unable to oxidise erythritol (Appendix A).

### 2.2. Genetic Diversity and Phylogeny of Isolated Strains

#### 2.2.1. BOX Fingerprinting

To assess the diversity of isolates in the collection, genomic BOX-PCR fingerprinting was performed based on the primer BOXA1R (5′-CTACGGCAAGGCGACGCTGACG-3′) [21]. An array of amplification bands show the relative positioning of the conservative BOX elements through the genome and, thus, reflect the common ancestry of the strains [22]. The genomic fingerprints obtained showed that the strains were quite heterogenic in the positioning of characteristic elements within the chromosome. A total of 18 groups of fingerprints can be formed among 39 strains, where only a few could be grouped together. The dendrogram of BOX-PCR groups (Figure 2) clearly shows the distant positioning of reference strains originating from America (a clade identified with purple). The strains circulating in Russia may be subdivided into three major clades, where the one identified with blue contains most strains isolated from crop plants.

#### 2.2.2. Phylogeny

A more detailed understanding of the diversity of strains in the collection can be obtained by analysing the sequencing data. For this purpose, either sequencing of the 16S rRNA gene [23] or sequencing of multiple bacterial housekeeping genes (MLSA) [24] is usually applied. The former method does not provide a reliable definition of the *Curtobacterium* species [18], and the latter method is costly. Moreover, bioinformatic analysis has shown that the primers proposed in the literature for the marker fragment of the gyrase gene (*gyrB*) are not suitable for genotyping the population of *Curtobacterium*, as they are fully complementary for only one third of the *Curtobacterium* sp. genomes presented in the NCBI Genbank [18]. Therefore, the development of universal markers whose PCR amplification sequencing enables reliable species differentiation of *Curtobacterium* sp. isolates is an urgent task.

An analysis of phylogenetic trees constructed using about 90 single-copied housekeeping genes, which was previously used effectively for MLSA taxonomic analysis [25], indicated that a relatively short sequence belonging to the *alaS*, i.e., alanine synthase gene, carried phylogenetic signals strong enough to separate most of the known genomic groups described earlier [18]. A pair of primers, AS-F and AS-R, was used to perform further bioinformatic analyses.

To verify the correct functioning of the primers, they were tested on a set of type *Curtobacterium* strains of the genus. The PCR products of expected length (506 bp) were amplified for all strains used (Figure 3).

Next, fragments of this marker gene amplified from the genomes of selected representative strains of each BOX group were sequenced to assess the phylogenetic relationship of these strains.

The *alaS* phylogenetic tree (Figure 4) indicated that the strains analysed can be assigned to several groups. Strain C139, isolated from peas, is closely related to the type strains Cfpf LMG 3645, as well as to the cluster of strains isolated from soybean (C089–C091). Reference strains C106 and C133 (Cf VKM Ac-2884), as well as strains C108, C110, C113 and C123, isolated from weed plants, can be closely related to *Curtobacterium flaccumfaciens* pv. *oortii* CFBP 3400. The close relationship of Cf VKM Ac-2884 and *Curtobacterium flaccumfaciens* pv. *oortii* CFBP 3400 has been shown previously, using complex phylogenetic analysis, including a large set of conserved genes [18]. The remaining strains belong to three different clusters containing *C. flaccumfaciens* strains and unclassified *Curtobacterium* sp. It is worth noting that the strains regarded as “yellow” clade according to BOX-PCR fingerprinting were clustered together by *alaS* sequence, too (Figure 4). This group of strains includes phytopathogens infecting plant hosts that are unusual for Cff, except for strain C144. The current taxonomy of *Curtobacterium* sp. makes even a rough classification of these strains difficult, regardless of the type of analysis used.

#### 2.2.3. MALDI-TOF MS Analysis

The registration of mass-spectra provides a basis for the identification and classification of bacterial strains. Although the “whole cell” spectrum reflects just a small part of the bacterial proteome, it is more suitable for a quick definition of taxonomic position considering a comprehensive database of MALDI-TOF spectra. This approach defines the unique “fingerprint” that characterises a microorganism. The spectra are unique and reproducible for bacterial genera and sometimes can specify species and subspecies.

MS-profiling clearly defined all strains studied as *Curtobacterium flaccumfaciens*. Further subdivision till pathovars was, however, not achieved, because of the limited number of available standards. Nevertheless, it was possible to plot the dendrogram (Figure 5), and the close clustering of strains attributed to the “blue” clade of BOX-PCR fingerprinting is noticeable. On the other hand, the strains supposedly referred to as being members of *C. flaccumfaciens* pv. *oortii* by *alaS* sequencing are located in different clusters.

### 2.3. Plant Pathogenicity and Resistance to Bacteriocides

#### 2.3.1. Pathogenicity Tests

All but four strains tested (C043, C115, C118 and C137) produced characteristic symptoms on inoculated soybean and common bean leaves. After 5 days on soybean and 7 days on common bean, areas of tissue with chlorosis began to appear, gradually changing to a straw-coloured necrosis (Figure 6). By day 12 after inoculation, leaf lesions were maximal. Plants inoculated with sterile water and non-pathogenic strains developed no symptoms, while leaves infected with the reference strain C001 developed typical tan spot symptoms. The bacteria were re-isolated from the symptomatic leaves and identified as Cff by pathovar-specific PCR [19].

The strains were unequally distributed in terms of virulence, as shown in Figure 7 and Appendix A. The correlation coefficient between the virulence of the strains on soybean and common bean was 0.701. Strains C088 and C142, initially isolated from soybean, were the most virulent for this plant (with average leaf damage zone widths of 7.07 and 6.84 mm, respectively), and the lowest values were obtained for strains C091 and C133 (with values of 0.87 and 0.88 mm, respectively). Common bean inoculation resulted in the highest virulence values for strains C086, C088 and C089 originating from soybean (5.50, 5.24 and 6.78 mm, respectively) and the lowest values for strains C039 (sunflower), C108 (ground elder) and C119 (American beech) (0.96, 0.93 and 0.92 mm, respectively). No virulence for soybean and common bean was observed for strains isolated from tomato (C043), ground elder (C115), thistle (C118) and wheat (C137), indicating a high level of pathogen specialisation of the host plant. It is interesting that these strains demonstrated impaired pectinase activity (Appendix A).

#### 2.3.2. Susceptibility of Cff Strains to Copper Compounds

An analysis of MIC and MBC for copper dihydroxide in a liquid medium showed that the tested strains had significant differences (Appendix A). For 38 strains, the MIC value was below 390 ppm. No accepted quantitative assessment of *Curtobacterium* sp. resistance to copper compounds exists, so these strains were classified as being copper sensitive. In contrast, strain C116 was rated as being resistant to Cu(OH)_2_, with a MIC value of 781 ppm. This strain was isolated from ground elder *(Aegopódium podagrária*), along with some other strains (C108-110; C113, C115-117; C123 and C130). The strains C113 and C115-117 were grouped together by BOX-PCR (Figure 2) and *alaS* alanine synthase gene sequencing (Figure 4). Strain C116, however, was separated from these groups on the basis of MALDI TOF analysis (Figure 5). Interestingly, the MIC values for two reference strains, C120 and C133, isolated from wild plants in America were twice as low as those of the majority of the strains (195 and 390, respectively). The reasons for the low/high sensitivity to the copper compound remain unclear.

## 3. Discussion

During the summers of 2018–2022, 33 strains of *Curtobacterium* sp. were isolated from symptomatic cultivated and wild plants in several regions of Russia. For each strain, taxonomic affiliation was confirmed by several molecular biological methods, including pathovar- and genus-specific PCR, the sequencing of the alanine synthase gene, BOX-PCR, MALDI-TOF and further phytopathological analysis.

Morphologically and biochemically, all the strains studied were highly uniform. All bacteria formed yellow, or light-yellow, colonies on NBY and SSM media, and revealed zones of hydrolysis on MSCFF medium. All strains were biochemically similar to the profile described in EPPO_Bulletin PM 7/100 [20].

Diagnostic PCR assays identified all the strains as Cff, a legume-specific pathovar of the pathogen *C. flaccumfaciens*. However, an in-depth genetic analysis revealed that each host plant can be infected with a noticeably wide range of the bacteria. This observation may contradict the pathovar definition, which associates the strain group with a particular limited set of host plants. Based on the results of the BOX-PCR fingerprinting method, the following conclusions can be drawn: all isolates acquired from soybean in 2021 were apparently close to one another and all had an identical BOX-PCR fingerprint (group III), despite the fact that they were isolated from soybeans grown throughout a fairly wide geographical range (Moscow, Kursk, Krasnodar, Novosibirsk regions).

These isolates form a cluster on the dendrogram (Figure 2), together with an isolate obtained from pea in 2022 (group XVI), an isolate obtained from apple tree seedlings in 2022 and a number of isolates obtained from wild *Sonchus* sp (VII).

The strains isolated from sunflower in the period 2018–2021 (group II) were also homogeneous in terms of their BOX-PCR fingerprints and were similar in pattern to other isolates from cultivated plants.

Conversely, the strains isolated from wild plants, even those growing in the same region, were distinct in terms of their high level of genetic diversity, as demonstrated by their BOX-PCR fingerprints and MALDI TOF profiles. Thus, isolates obtained from *Aegopodium* sp. in the Moscow region could both be grouped with the classic reference strain C001, forming a separate clade, and fall into all other clades, with the exception of the outer group formed by the majority of reference strains (groups XIII, X, IX).

The results of phylogenetic analysis based on *alaS* gene sequencing showed the similarity of the type strain of *C. flaccumfaciens* pv. *flaccumfaciens* LMG 3645 with a few of the analysed strains. Interestingly, most of the strains were closer to C. *flaccumfaciens* pv. *oortii* CFBP 3400, although consistent classification of representatives of the genus *Curtobacterium* requires, first of all, the development of a classification scheme that takes into account all known genomic data. This will allow for a clear classification based on whole-genome sequencing. The *alaS* phylogenetic tree partially replicates the topology of the tree obtained from BOX-PCR fingerprints, but there are some differences. The same can be said about MALDI TOF clustering. In all three cases, the 2021 soybean isolates clustered with the 2020 wild plant isolates. Sunflower-originated strains also grouped together, forming a common clade with representatives of the BOX groups V, I and XI.

Based on the data obtained, it can be assumed that infected seeds provide the major means of distribution of *C. flaccumfaciens* strains, leading to little genetic diversity among isolates obtained from cultivated plants growing in a broad geographical area. In addition, the fact that similar strains were isolated from both cultivated and wild plants may indicate that wild plants can be a significant reservoir of potential infection.

The high degree of correlation (0.701) between strains’ aggressiveness towards soybean and common bean may indicate that the mechanisms responsible for virulence provide effective virulence for different crops, including when they were initially isolated from non-host plants. The results obtained are consistent with those reported by Harveson, 2015 and Gonçalves, 2017 [7,26], where the strains isolated from barley (*Hordeum vulgare*), black and common (or white) oat (*Avena strigose* and *sativa*), rapeseed (*Brassica napus*), ryegrass (*Lolium* spp.), wheat (*Triticum* spp.), tomatoes (*Solanum melongena* and *lycopersicum*) and pepper (*Capsicum* spp.) [27] were found to be pathogenic for leguminous crops, in particular for common bean. Differences in virulence observed during inoculation by different strains may be associated with variability of the complex system of regulation of virulence factors. For example, it has been shown that strains isolated from common bean were weakly virulent for soybean, but that inoculation with the same strains re-isolated from soybean showed strong virulence against soybean [28].

Conversely, when developing methods for pathogen control, it is necessary to differentiate between common bean, yellow melilot and soybean in crop rotation, since most strains can affect all of these crops, according to the data obtained. As an additional control method, it is necessary to combat weed plants, such as thistle species (*Sonchus* sp.), ground elder (*Aegopódium podagrária*) and many others [17]. Currently, it is known that Cff is capable of infecting and colonising a wide range of plants [5], and this list of potential plant hosts is incomplete. Some additional research may be required to achieve a comprehensive view of the biology of the pathogenesis of the bacterium on the plant.

The strains also differed in their susceptibility to copper dihydroxide. For example, strain C116, in spite of having been collected from a weed plant, demonstrated enhanced resistance to Cu(OH)_2_. It is possible that this strain was exposed to treatments with copper-based fungicides, to protect the crop plants from the diseases, which provided resistance to increased doses of copper. Although not critical, this is a serious issue, since the emergence of copper-resistant forms of the pathogen can reduce the effectiveness of treatment against the disease, because copper-based fungicides are used to control Cff on plants [29]. On the other hand, the resulting resistance of the strain is interesting to study, since precedents for the emergence of copper-resistant Cff strains are unknown, in spite of this being a common phenomenon in other phytopathogenic bacteria [30].

## 4. Materials and Methods

### 4.1. Bacterial Strains: Isolation and Growth Conditions

There were 39 strains of *Curtobacterium* sp. studied. The type strain of Cff and five other reference strains were acquired from the All-Russian Collection of Microorganisms (VKM). Thirty-three novel strains were isolated from soybean and other plants with the symptoms of bacterial spots, blight and wilting (Table 1).

To isolate pathogenic bacteria, an Erlenmeyer flask containing 100 g of symptomatic tissue (mostly leaves and stems) was filled with 300 mL of sterile physiological solution (SPS; 8.5 g NaCl, 991.5 mL distilled water) and agitated for 12 h at 200 rpm and at 4 °C [31].

Then, 2 mL volume of the extract was precipitated at 7000 rpm for 20 min at 4 °C (Eppendorf 5430). The supernatant was decanted and the pellet was homogenised in 1.5 mL SPS. Then, 100 μL of the suspension with five ten-fold dilutions in SPS was plated onto MSCFF medium (per 1 L: peptone—5 g; meat extract—3 g; sucrose—5 g; agar—15 g; skimmed milk powder—5 g; Congo red (Dia-M, Moscow, Russia)—0.05 g; chlorothalonil * (Bravo, SC, Syngenta, Basel, Switzerland)—0.01 g; thiophanate methyl * (Topsin-M, KS, Nippon Soda, Tokyo, Japan)—0.01 g; nalidixic acid * (Dia-M, Moscow, Russia)—0.01 g; nitrofurantoin * (Sigma, St. Louis, MO, USA)—0.01 g; oxacillin sodium salt * (Sigma)—0.001 g; sodium azide—0.001 g); (*—added after autoclaving the medium) [32]. In the selection of typical colonies, the research was guided by the reference strain C001, which has the following characteristics: the colour of the colonies is yellow, the consistency is mucoid, there is a transparent zone of 7–10 mm around the colonies on the MSCFF medium, as a sign of hydrolysis of casein and Congo red.

The strains were cultured for 7 days at 28 °C. Growth rate was then assessed by the appearance of colonies as compared with the growth of the reference strain. Colonies grown on this medium were transferred to SSM medium (per 1 L: rhamnose—5.0 g; yeast extract—2.0 g; KH_2_PO_4_—0.5 g; K_2_HPO_4_—2.0 g; NH_4_Cl—1.0 g; LiCl—10.0 g; MgSO_4_·7H_2_O *—0.25 g; Tris-HCl—1.2 g; sodium azide *—2.0 g; cycloheximide * (Sigma)—0.1 g; polymyxin sulphate B * (Sigma)—0.04 g; bromocresol purple (Dia-M) * (15% solution in ethanol)—1 mL; agar 15 g; (*—added after autoclaving the medium) [33].

Further routine cultivation was performed using YD medium (yeast extract—10 g/L; dextrose—20 g/L; agar—15 g/L) at 28 °C for 24 h. Strain stocks were stored in 15% glycerol at −80 °C. Detailed information on the strains used can be found in Table 1.

### 4.2. DNA Extraction

DNA was isolated using the GeneJET Genomic DNA Purification Kit (Thermo Fisher Scientific, Waltham, MA, USA), according to the Gram-Positive Bacteria Genomic DNA Purification Protocol. The quality and quantity of the extracted DNA was measured using a NanoPhotometer N60 spectrophotometer (Implen, Westlake Village, CA, USA). DNA was stored at −20 °C.

### 4.3. Molecular Identification of Isolated Strains

Primary PCR molecular identification was performed using a set of genus-specific primers Curto-F2 5′-GAAATGGTGTTATGGCCGGAT-3′ and Curto-D-R 5′-ACGGGTTAACCTCGCCACA-3′, according to the protocol recommended in [18]. Expected PCR product with these primers was ~275 bp.

The strains were additionally PCR tested with a set of primers recommended for the detection of Cff CffFOR2 5′-GTTATGACTGAACTTCACTCC-3′ and CffREV4 5′-GATGTTCCCGGTGTTCGA-3′ [19]. Expected PCR product with these primers was ~305 bp.

The reaction mixture with a volume of 25 µL contained 5 µL 5× ScreenMix (Evrogen, Moscow, Russia), 0.5 µM of each primer, 18 µL Milli-Q water and 1 µL (50 ng) DNA. PCR conditions were initial denaturation at 94 °C for 3 min, followed by 30 cycles of denaturation at 94 °C for 1 min; annealing at 62 °C for 45 s, elongation at 72 °C for 30 s and final elongation at 72 °C for 5 min. Amplicons were separated by electrophoresis in a 1% agarose gel in TAE buffer containing ethidium bromide.

The presence of a PCR product of the expected length was considered to be a positive signal.

### 4.4. Genetic Fingerprinting

BOX-PCR was used to evaluate the genetic diversity of the isolated strains, according to EPPO standard PM 7/100 [20]. A total of 35 µL of the reaction mixture contained 3.5 µL 10× Turbo buffer (Evrogen, Moscow, Russia), 1.4 µL dNTP, 2.1 µL BOXAIR primer (5′-CTACGGCAAGGCGACGCTGACG-3′, 10 mM), 0.3 µL Taq DNA polymerase, 26.7 µL Milli-Q water and 1 µL (50 ng) DNA. The following PCR parameters were used: 95 °C for 10 min, then 34 cycles of 95 °C for 1 min, 52 °C for 1 min and 72 °C for 1 min.

The amplified fragments were analysed by capillary electrophoresis, using a QIAxcel Advanced System (Qiagen, Hilden, Germany). For the experiment, a corresponding DNA High Resolution Gel Cartridge was used in conjunction with the OM500 method. Size marker 100 bp–2.5 kb (20 ng per µL) and alignment marker QX 15 bp–3 kb were used to calculate the length of the PCR product.

A dendrogram derived from BOX-PCR fingerprints was created with PyElph 1.4, software used for gel image analysis and phylogenetics [34].

### 4.5. Sequencing and Analysis of Alanine Synthetase

PCR amplification of the alanine synthase gene (*alaS*) was performed using the primer set AS-F (5′-TTCCAGATGAACGGBAACTTC-3′) and AS-R (5′-TGGTCGRTCTCGTACATGTTG-3′), which was developed specifically for this study and optimised for *Curtobacterium* phylogenetics. The composition of the PCR mixture for each 25 µL reaction included 5 µL 5× ScreenMix (Evrogen, Moscow, Russia), 18.8 µL Milli-Q water, 1 µL (50 ng) DNA and 0.1 µL (100 µM) of each primer. The purity and yield of PCR products were verified by electrophoresis of the reaction product on a 1% agarose gel stained with ethidium bromide.

The bands of PCR products were excised from the agarose gel and DNA was purified using a QIAquick PCR Purification kit (Qiagen, Hilden, Germany), following the manufacturer’s protocol. Sanger sequencing in both directions was carried out by Evrogen (Moscow, Russia).

### 4.6. Phylogenetic Analysis

Alignments of *alaS* PCR amplicon sequences were obtained using MAFFT 7.48 with AUTO settings [35]. Phylogenetic trees were constructed using IQ-TREE 2.2.2.7 with “--alrt 1000 -B 1000” parameters [36]. Ultrafast bootstrapping (1000) was used to assess the robustness of trees. The *alaS* phylogenetic tree was visualised using iTOL v6 [37].

### 4.7. MALDI-TOF

For matrix-assisted laser desorption/ionisation time-of-flight mass spectrometry (MALDI-TOF MS), bacteria were grown on R2A medium (Thermo Fisher, Maltham, MA, USA) at 28 °C for 96 h (with four independent experiments for each strain). The samples were prepared from fresh colonies, as described previously [38], and analysed using an Autoflex Speed mass spectrometer (Bruker Daltonics, Bremen, Germany), according to the manufacturer’s recommendations. Analysis of the mass-spectra was performed using the spectrum view of Flex analysis 3.3 and MALDI Biotyper 3.0 software (Bruker Daltonics).

### 4.8. Biochemical Characterisation of Curtobacterium Strains

Various physiological and biochemical tests were performed to investigate the phenotypic characteristics of bacteria of the genus *Curtobacterium*, as recommended by EPPO standard 7/102 (https://www.eppo.int, accessed on 1 November 2023). In this study, the morphology of bacterial colonies was examined on NBY medium [39] and semi-selective MSCFF medium [32].

Gram staining was checked using both a classical staining procedure and a rapid test with 3% KOH.

Catalase activity was determined by adding one drop of 3% H_2_O_2_ to one drop of the strain suspension. In the case of a “bubbling” reaction, the result was regarded as positive.

Oxidase activity, urease activity, indole formation and O/f tests were performed with the “Paper Indicator System for Identification of Microorganisms, Set #1” (Microgen, Moscow, Russia), according to the manufacturer’s protocol.

A gelatin medium at room temperature was used to determine gelatinase activity (meat extract—3 g/L; peptone—5 g/L; gelatin—120 g/L). The medium was poured into test tubes and the biomass was introduced by stabbing the microbial loop into the medium. Then, the tubes were incubated at 28 °C for 48 h and, after cooling the medium, the presence and degree of liquefaction were determined [40].

Phenol Red Maltose Broth (peptone 10—g/L; sodium chloride—5 g/L; maltose 5 g/L; phenol red—0.018 g/L) was used to evaluate maltose oxidation. Acidification of the medium was assessed by the change in colour of the indicator to yellow. For the oxidation of inulin and erythritol oxidation, the same approach was used with the corresponding carbohydrate in the medium.

### 4.9. Pathogenicity Tests

Experiments featuring the artificial infection of soybean and common bean plants were conducted from May to September 2023, in the RSAU-MTAA’s experimental greenhouses.

Bacterial strains were grown on YD agar medium (YDC without CaCO_3_) at 28 °C for 72 h, according to [41]. Bacterial suspensions were prepared by adding 10 mL of sterile distilled water to each Petri dish and adjusting the concentration to 10^8^ CFU/mL, and were assessed spectrophotometrically at 600 nm. For better contact with the leaf, the wetting agent Silwet Gold (Chemtura, Philadelphia, PA, USA) was added to the suspension at a concentration of 0.01%. Sterile distilled water with a wetting agent and a suspension of strain C001 were used as negative and positive controls, respectively.

Prior to infection, soybean cv. Kasatka was grown in a peat–perlite substrate in plastic trays until stage V3 (plants had four nodes with three fully expanded trifoliate leaves). Seedlings were grown in a greenhouse at 28/22 °C (14 h day/10 h night) with natural light and irrigated as needed. Two days before, and 24 h after, inoculation, a relative humidity of ~95% was maintained at 28/22 °C.

The pathogenicity of the strains was tested by cutting the leaves with scissors soaked in a bacterial suspension, perpendicular to the veins, to a depth of 1 cm from the leaf margin, according to [42]. All the leaves of each plant were thus infected. Three replicates with three plants each were analysed. Symptoms were recorded on the 12th day after inoculation, by measuring the width of the leaf necrosis zone with a calliper, for all leaves of all plants, and calculating the average value of the indicator for the stem. The common bean plants cv. Purpurnaya Koroleva were grown in peat–perlite substrate in plastic trays until stage V2. The other parameters for growth and inoculation were the same as for soybean. Each experiment was repeated twice: in May–June and in August–September 2023.

To complete the Koch triad in confirming the pathogenicity of the strains, the bacteria were re-isolated from leaves with symptoms and identified as Cff by PCR [19], according to [43], with modifications. For this purpose, the symptomatic tissue area was surface sterilised in 70% ethanol, washed five times with sterile water, placed in 0.5× buffer overnight at 4 °C, homogenised with a sterile laboratory pestle, filtered through a cheesecloth and centrifuged for 15 min at 8000 rpm in an Eppendorf 5430 centrifuge (Eppendorf, Hamburg, Germany). Bacterial DNA was isolated using a Cytosorb DNA extraction kit (Syntol, Moscow, Russia), according to the manufacturer’s protocol. The isolated DNA was used directly for PCR and confirmation of re-extraction of the pathogen from the affected leaves.

### 4.10. Susceptibility of Bacterial Strains to Copper Hydroxide

The determination of the in vitro sensitivity of the studied strains to Cu(OH)_2_ was carried out according to the method described in [44], with modifications. A commercial product based on copper dihydroxide (350 g/kg) (Kocide^TM^ 2000, WDG, Corteva Agriscience, Indianapolis, IN, USA) was diluted in YD liquid broth medium (YDC without CaCO_3_ and agar) in a 1:1 ratio (14 dilutions).

Assay cultures were prepared by resuspending individual colonies in 5 mL YD broth and incubating for 24 h at 28 °C and at 150 rpm in an ES-20 shaker (BioSan, Riga, Latvia). The bacterial suspensions were diluted with liquid YD medium to a concentration of ≈10^5^ CFU/mL based on OD_600_.

YD liquid medium (90 µL), drug dilutions (100 µL) and suspensions of the individual strains (10 µL) were added to sterile 2 mL Eppendorf tubes. The total volume of the culture mixture was 200 µL. The concentrations tested were 50,000, 25,000, 12,500, 6250, 3125, 1562.5, 781.25, 390.6, 195.3, 97.6, 48.8, 24.4, 12.2 and 6.1 ppm (in terms of a.i.). Tubes containing 200 µL of liquid YD medium (without agar) were used as a negative control for each replication, and a tube containing 190 µL of liquid YD medium and 10 µL of suspension of each strain was used as a positive control. The tubes were then thoroughly mixed on a vortex shaker and incubated for 48 h at 28 °C and 350 rpm in a ThermoMixer F 2.0 incubator (Eppendorf, Hamburg, Germany).

After 48 h of incubation, 10 µL of the mixture from each tube was diluted ten-fold with sterile water, in separate 2 mL tubes. A total of 100 µL of each dilution was surface spread on YD agar medium and cultivated at 28 °C for 48 h. The growth of the bacteria was then visualised and the concentration of bacteria in each tube was calculated. The minimum inhibitory concentration (MIC) was defined as the lowest concentration of Cu(OH)_2_ that resulted in 90% inhibition of growth, compared with the control. The minimum bactericidal concentration (MBC) was defined as the lowest concentration of Cu(OH)_2_ that killed 99.9% of bacteria. The tests were performed with four replicates on each strain. Strains with a MIC ≤ 390 ppm were considered sensitive to copper dihydroxide.

### 4.11. Statistical Analysis

Data analysis was performed by analysis of variance with Statistica 12.0 software (StatSoft, Tulsa, OK, USA); the comparison of means was carried out using Duncan’s test. *p*-values < 0.05 were considered significant.

## Figures and Tables

**Figure 1 plants-13-00667-f001:**
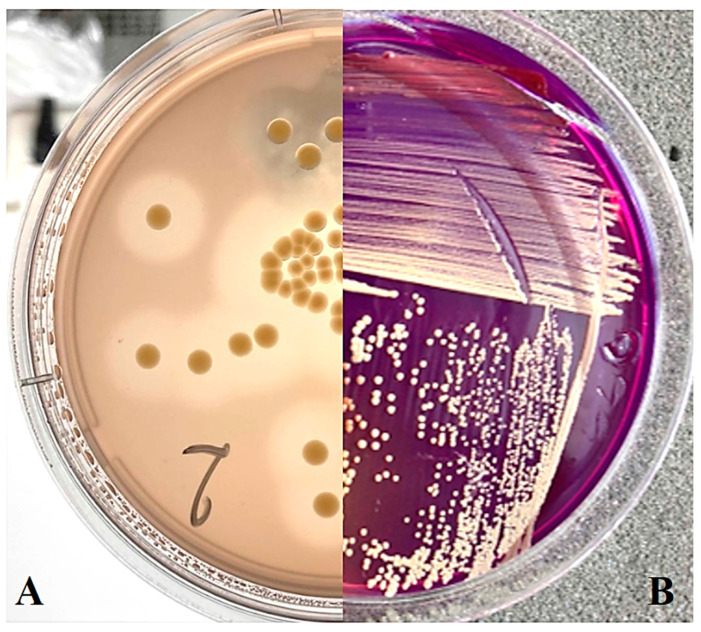
*Curtobacterium* sp. colonies on nutrient media: (**A**) MSCFF on the 5th day of cultivation; (**B**) SSM on the 12th day of cultivation.

**Figure 2 plants-13-00667-f002:**
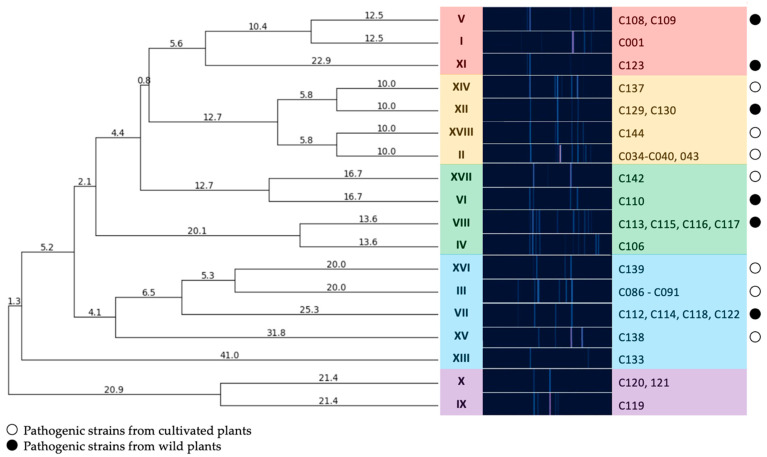
A dendrogram constructed according to BOX-PCR fingerprints of Cff strains (UPGMA method). Clades are highlighted using different colours. PyElph 1.4 software was used for dendrogram construction and to calculate the genetic distances between strains.

**Figure 3 plants-13-00667-f003:**
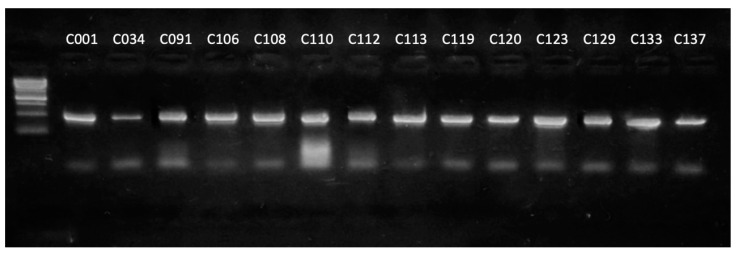
Results of the electrophoretic separation of the amplification products of the *alaS* gene using strains of the genus *Curtobacterium* as a DNA template. Marker—1 kb DNA ladder (Evrogen).

**Figure 4 plants-13-00667-f004:**
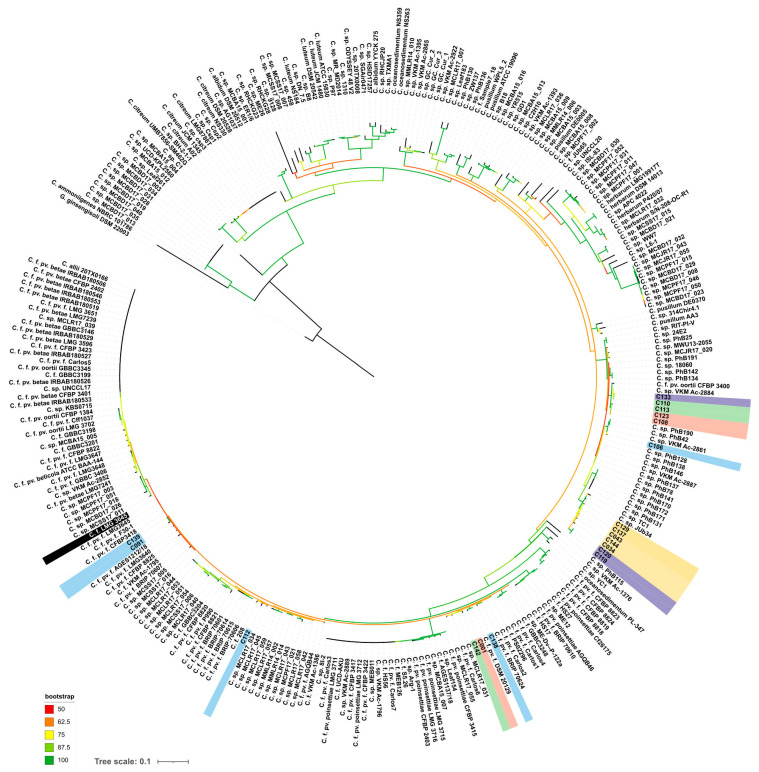
Phylogenetic tree based on nucleotide sequences of alanyl-tRNA synthetase. Studied strains are shown with a background coloured in the same scheme as in Figure 2. Type strain Cff LMG 3645 is shown with a black background. *Gryllotalpicola ginsengisoli* DSM 22003 was used as an outgroup. The scale bar shows 0.1 estimated substitutions per site. The total number of bootstrap trees was 1000. Branches with bootstrap support of 50% and higher are shown using the colour gradient explained in the legend.

**Figure 5 plants-13-00667-f005:**
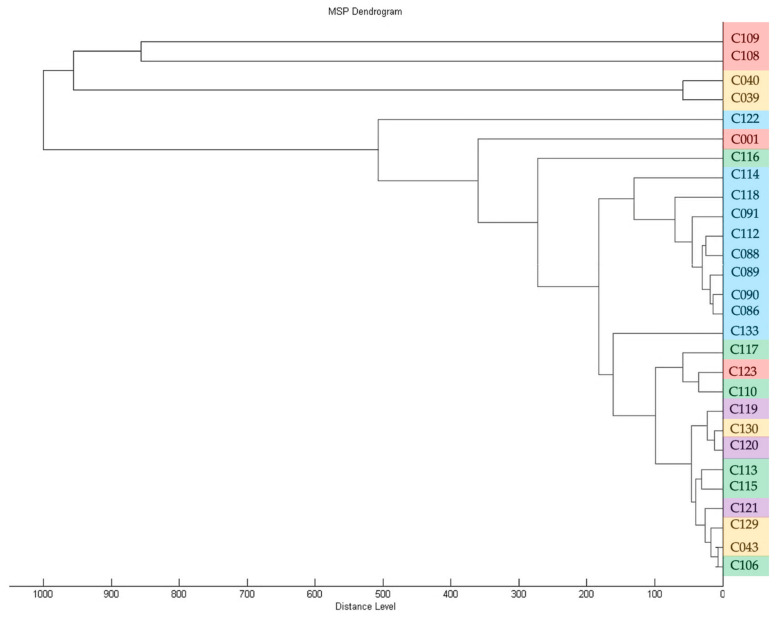
Dendrogram derived from MALDI-TOF MS protein profiling of strains used in this study, generated with Biotyper 3.0 (Bruker Daltonics). The colour coding of strain numbers matches that used in Figure 2.

**Figure 6 plants-13-00667-f006:**
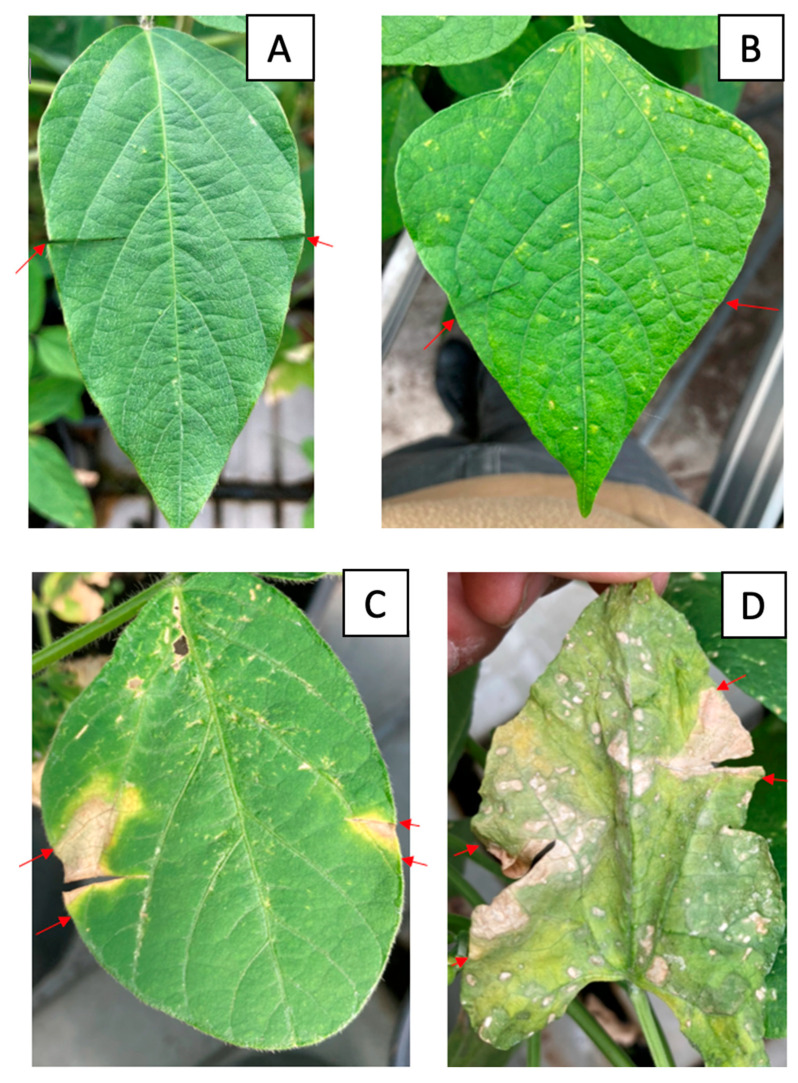
Symptoms of infection with Cff strains on soybean and common bean leaves. (**A**) Negative control on soybean, (**B**) negative control on common bean, (**C**) inoculation with strain C089 on soybean leaves, (**D**) inoculation with strain C086 on common bean leaves. The red arrows indicate the boundaries of the infected area of the leaf.

**Figure 7 plants-13-00667-f007:**
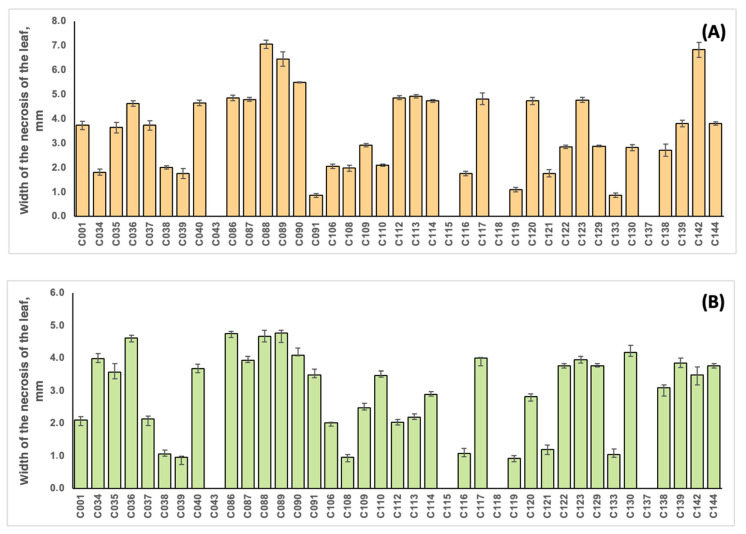
The width of the leaf damage zone during the inoculation of soybean plants cv. Kasatka (**A**) and common beans cv. Purpurnaya Koroleva (**B**) with different strains of Cff. The values represent the respective mean of three independent trials; error bars represent the standard deviation.

**Table 1 plants-13-00667-t001:** List of strains used in this study.

No	Strain	Name	Origin	Year
**Pathogenic Strains from Cultivated Plants**				
1	C034	SF20	Sunflower, Kursk region	2018
2	C035	SF21		2018
3	C036	SF22		2018
4	C037	SF23		2018
5	C038	SF24		2018
6	C039	SF25		2021
7	C040	SF26		2021
8	C043	Curt3	Tomato, Moscow region	2021
9	C086	F125-1	Soybean, cv. Kasatka, Ryazan region	2021
10	C087	F125-2		2021
11	C088	F125-3		2021
12	C089	F30-1	Soybean, cv. Dauria, Amur Region	2021
13	C090	F30-2		2021
14	C091	F30-3		2021
15	C137	362	Wheat, Moscow region	2022
16	C138	429	Apple tree seedlings, Moscow region	2022
17	C139	507	Pea, Moscow region	2022
18	C142	44	Soybean cv. Nordica, Belgorod region	2022
19	C144	19	Soybean, cv. Sultana, Novosibirsk region	2022
**Strains from Wild Plants**				
20	C108	414DL	Ground elder (*Aegopódium podagrária*), Moscow region	2020
21	C109	557DL	*A. podagrária*, Moscow region	2020
22	C110	412DL		2020
23	C112	329DL	Sow thistle (*Sonchus* sp.), Moscow region	2020
24	C113	415DL	*A. podagrária*, Moscow region	2020
25	C114	367DL	*Sonchus* sp., Moscow region	2020
26	C115	411DL	*A. podagrária*, Moscow region	2020
27	C116	575DL		2020
28	C117	571DL		2020
29	C118	332DL	*Sonchus* sp., Moscow region	2020
30	C122	144DL		2020
31	C123	570DL	*A. podagrária*, Moscow region	2020
32	C129	53150	Yellow melilot (*Melilótus officinális*), Krasnodar region	2019
33	C130	400DL	*A. podagrária*, Moscow region	2020
**Reference Strains**				
34	C001	Ac-1923 (DSM 20129 ATCC 6887, NCTC 4758)	Common bean (*Phaseolus vulgaris*)	before 1990
35	C106	52862 = VKM Ac-2861	*Marah* sp., California, USA	2019
36	C119	53223	American beech (*Fagus grandifolia*), New York, USA	2019
37	C120	53217	*F. grandifolia*, New York, USA	2019
38	C121	53256	*Agrostis capillaris*, nematode *Anguina agrostis*, Washington, USA	2020
39	C133	53258 = VKM Ac-2884		2020

## Data Availability

Data are contained within the article and Appendix A.

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
