# Peer review of "Phytopathogenic Curtobacterium flaccumfaciens Strains Circulating on Leguminous Plants, Alternative Hosts and Weeds in Russia"

_plants, 2024, doi:10.3390/plants13050667_

Round 1

Reviewer 1 Report

Comments and Suggestions for Authors

Dear Authors

 in the attached file you can find my suggestions,

best regards

Reviewer 2 Report

Comments and Suggestions for Authors

Review report

Comments and suggestions to the authors

A brief summary

The manuscript “Phytopathogenic Curtobacterium flaccumfaciens strains circulating on leguminous plants, alternative hosts and weeds in Russia” emphasizes on isolation and characterization of actinobacterial strains, isolated from different plant origins in Russia. According to a report by EPPO Curtobacterium flaccumfaciens pv. flaccumfaciens is considered a pathogen of high-risk and is included in the A2 list of quarantine pests. Among most affected countries several could be mentioned: Canada, Еastern Australia, Iran, central Plants of the America and Brazil. The situation in Europe is different. According to the EFSA report from 2018, the presence of Cff has been documented sporadically in several countries but at the time of data collection (year 2018), no active occurrence of the infection has been reported on the continent. The study of the dispersal of the pathogen in Europe and the collection of new data is crucial. In this regard, the manuscript is valuable and interesting and should be considered for publication. However, some questions arose during the evaluation of the paper.

Abstract: clearly written;

Introduction: the information is clear and fully consistent with the main topic of the manuscript. Several minor remarks are listed below:

Line 50 – 56: A relevant reference is needed here.

Line 61 – 62: “Similar measures have also been taken by the Inter-African Plant Protection Council and the Caribbean Plant Protection Commission” – a relevant reference is needed here.

Line 70: Please, clarify which EPPO report you have cited here? The reference 9 is the EFSA report.

Line 76 – 78: This sentence is unclear. Please, rearrange it in order to become clearer.

Materials and methods: The section is good written. However, several minor remarks are listed below:

Subsection 4.1.

Line 317: Please, specify here how many are the new isolates and how many are the reference strains used in the study;

Line 321:” symptomatic tissue” – please, specify here from which plants’ parts (leaves, stems, petioles etc.)?

Line 322: give the full name of STS;

Line 341: MgSO4x7H2O, subscript

Subsection 4.3. The whole section should be rewritten in order to become clearer.

I suggest the expected PCR product (lengths in bp) to be given for both PCR reactions.

Line 361 – 366: Please, clarify here for which PCR reaction are the given conditions: genus or species - specific?

Line 362: “50 ng DNA” but in what volume. It is more correct to unify the way you give this information. Below, the DNA quantity is given in µL, without information for the concentration (line 374).

Line 361 – 362: give the volume of the water;

Subsection 4.4.

Line 371 – “EPPO standard PM 7/100” – a relevant reference is needed here;

Subsection 4.5.

Line 388 – 390: recalculate the amount of the reagents in this PCR reaction. The final volume is not 25 µL.

Line 390: I suggest here to give the length of the PCR product;

Subsection 4.8.

Line 418: it is correct to write hydrogen peroxide or H2O2, not just peroxide.

Subsection 4.9.

Line 446: please, explain what stage V3 means.

Subsection 4.10

Line 474: I suggest here to write YD broth, instead “with no agar”;

Line 488 – 490: You can shorten the text here. I suggest the following: “One hundred µL of each dilution were surface spread on YD agar medium and cultivated at 280C for 48 h.” It is not good to start the sentence with a number.

Results: the section is clear with several minor suggestions listed below:

Line 85 – Please, clarify here in the text that these 33 strains are your isolates and the remaining 6 are the reference strains.

Line 93 – NBY medium – Figure 1 represent MSCFF and SSM media. Please, clarify here.

Line 105 – 106: This sentence is unclear. It is not clear if all isolated strains were identified as Cff? Please, clarify here.

Subsection 2.2.3 MALDI analyses. If you mean MALDI-TOF MS analyses you should write it. Please, clarify here.

Line 176 – 182: This section is unclear. “Although the “whole cell” spectrum reflects just a small part of the bacterial proteome, it is more suitable for a quick definition of taxonomic position than a comprehensive database of MALDI-TOF spectra” – you should be more specific here.

 “This approach defines the unique “fingerprint” that characterizes a microorganism. The spectra are unique and reproducible for bacterial genera and sometimes can specify species and subspecies.” I suppose that this statement refers to MALDI-TOF? Or to MALDI as you wrote? Please, clarify here what is the difference (if any) between MALDI-TOF and MALDI and which analysis is the more reliable approach, according to you.

Discussion: this section is well written and informative.

Line 287: “The results obtained are consistent with those reported by………[5,23]………..”
